# Interaction of DDB1 with NBS1 in a DNA Damage Checkpoint Pathway

**DOI:** 10.3390/ijms252313097

**Published:** 2024-12-05

**Authors:** Hoe Eun Lim, Hee Jung Lim, Hae Yong Yoo

**Affiliations:** 1Department of Health Sciences and Technology, Samsung Advanced Institute for Health Sciences and Technology, Sungkyunkwan University, Seoul 06351, Republic of Korea; imhoeeun@skku.edu; 2Research Institute for Future Medicine, Samsung Medical Center, Seoul 06351, Republic of Korea

**Keywords:** NBS1, DDB1, MDC1, TopBP1, DNA damage checkpoint, Xenopus egg extract

## Abstract

Various DNA damage checkpoint control mechanisms in eukaryotic cells help maintain genomic integrity. Among these, NBS1, a key component of the MRE11-RAD50-NBS1 (MRN) complex, is an essential protein involved in the DNA damage response (DDR). In this study, we discovered that DNA damage-binding protein 1 (DDB1) interacts with NBS1. DDB1 is a DDR sensor protein found in UV-induced DNA replication blocks. Through pull-down and immunoprecipitation assays conducted in Xenopus egg extracts and human cell lines, we demonstrated a specific interaction between NBS1 and DDB1. DDB1 was also found to associate with several proteins that interact with NBS1, including DNA topoisomerase 2-binding protein 1 (TopBP1) and Mediator of DNA damage checkpoint protein 1 (MDC1). Notably, the interaction between DDB1 and NBS1 is disrupted in MDC1-depleted egg extracts, indicating that MDC1 is necessary for this interaction. Furthermore, the depletion of DDB1 leads to increased Chk1 activation upon DNA damage. These novel findings regarding the interaction between NBS1 and DDB1 provide new insights into how DDB1 regulates DNA damage pathways.

## 1. Introduction

In eukaryotic cells, mechanisms known as damage checkpoints are crucial for maintaining the integrity of the genome. When DNA damage occurs, it activates signaling pathways that regulate crucial processes such as cell cycle checkpoints and DNA repair [1,2]. The detection of DNA damage involves the sensing of the damage, leading to the activation of two phosphoinositide 3-kinase-related kinases (PIKKs), namely Ataxia Telangiectasia Mutated (ATM) and Ataxia Telangiectasia and Rad3 related (ATR). These kinases phosphorylate a variety of proteins across different specialized pathways [3,4]. ATM is chiefly responsible for the initial detection of double-stranded breaks (DSBs), which are considered extremely harmful forms of genetic damage, typically resulting from exposure to ionizing radiation (IR) or certain genotoxic agents [5]. Conversely, ATR becomes active in situations involving replication stress or blockage [3,4,6]. Remarkably, ATR also plays a role in the response to DSBs by working together with ATM [7,8,9].

ATR activates the checkpoint kinase Chk1 when replication forks encounter stalling [10,11]. The active form of Chk1 phosphorylates and regulates the function of various proteins, including significant regulators of the cell cycle, such as Cdc25 and Wee1 [12]. The activation of ATR is tightly controlled through its interactions with multiple proteins. One key player is ATR-interacting protein (ATRIP), which is essential for the localization of ATR to DNA damage sites and for facilitating its kinase activity [13]. Further investigations have revealed that TopBP1 plays a role in activating the ATR-ATRIP complex [14]. ATRIP is also necessary for linking ATR to TopBP1 [15]. In the context of double-stranded breaks (DSBs), ATR becomes activated through the phosphorylation of TopBP1 by ATM, specifically at the S1131 residue, significantly enhancing its ability to activate ATR. This makes TopBP1 a key substrate regulated by ATM in checkpoint responses [9].

In earlier reports, we highlighted that the activation of ATR in response to DSBs involves the interaction between TopBP1 and NBS1, a member of the MRN complex within Xenopus egg extracts [16]. Recent research has shown that TopBP1 forms an association with NBS1 in human cells [17]. The MRN complex serves as a primary sensor for DSBs, playing a crucial role in attracting ATM to sites of DNA damage. Upon reaching a DSB, ATM directly interacts with NBS1 [18].

In egg extracts depleted of NBS1, ATM fails to bind to TopBP1, highlighting the crucial role these interactions play in recruiting ATM to TopBP1. This suggests that the MRN complex facilitates the connection between ATM and TopBP1 [16]. In human cells, the positioning of the MRN complex to IR-induced foci relies on the presence of MDC1 [19]. MDC1, an adaptor protein, plays a significant role in DDR by localizing to DNA damage sites through its binding to γ-H2AX. Additionally, MDC1 interacts specifically with NBS1 [20,21,22,23,24]. Our recent findings identified MDC1 as a protein that interacts with both TopBP1 and NBS1 in egg extracts. Furthermore, MDC1 is necessary for linking TopBP1 to NBS1, both in egg extracts and in human cells. MDC1 regulates the DDR by interacting with the BRCT I-II domain of TopBP1 using its amino acid region 161–230 and its deletion indicates a failure to phosphorylate Chk1 [24]. This interaction is crucial for ATM to facilitate ATR-ATRIP activation in response to DSBs.

NBS1 is pivotal in the DDR due to its interactions with various other proteins. Our goal was to explore the function and regulation of NBS1 further. To achieve this, we examined proteins that interact with NBS1 in Xenopus egg extracts. We carried out pull-down experiments to isolate potential interacting candidates, followed by mass spectrometry analysis. Through this process, we found DNA damage-binding protein 1 (DDB1) to be a noteworthy NBS1-interacting candidate. DDB1 is a multifunctional protein recognized as part of a heterodimeric complex responsible for detecting DNA lesions resulting from UV damage through the nucleotide excision repair pathway [25]. Additionally, DDB1 is a key component of the ubiquitin–E3 ligase complex, functioning as an adapter between Cullin 4A and CUL4-associated factors, facilitating the ubiquitination of substrates. The Cul4A/DDB1 complex plays a significant role in checkpoint recovery by promoting the proteolysis of Chk1 via ubiquitin-mediated mechanisms in response to replication stress and DNA damage [26]. Furthermore, DDB1 interacts with multiple proteins that are crucial for regulating DNA repair, cell cycle progression, and gene transcription [27].

This study identified DDB1 as a protein that interacts with NBS1 in both egg extracts and human cells. Additionally, DDB1 interacts with several other proteins that interact with NBS1, which are essential regulators of the checkpoint responses to DNA damage. Furthermore, MDC1 is necessary for facilitating the interactions between DDB1 and NBS1. Notably, the depletion of DDB1 in both egg extracts and human cells leads to an increased activation of Chk1 in response to DNA damage.

## 2. Results

### 2.1. NBS1 Interacts Specifically with DDB1

The MRN complex acts as DSB’s key mediator and sensor protein for ATM and plays an important role in replication block damage [28,29]. We focused on exploring the functions and interacting proteins associated with the MRN complex, with particular emphasis on NBS1. Identifying proteins that interact with NBS1 may provide insights into the regulators involved in DNA damage checkpoint responses. To find these interacting proteins, we conducted pull-down experiments utilizing a recombinant form of NBS1 that includes the FLAG epitope (NBS1-FLAG) derived from Xenopus egg extracts with annealed oligomers of dA_70_ and dT_70_ (referred to as pA-pT) (Figure 1A). This template is known to activate DNA damage checkpoints through mechanisms that rely on crucial upstream regulators such as ATM, ATR-IP, and TopBP1 [9,14,30]. Several bands with different patterns were observed after treatment with pA-pT. We divided the gel into five sections for mass spectrometry analysis, but we did not find any significant new proteins in the other regions; we only obtained results similar to those in Appendix A for the 90–150 kDa region (Figure 1A). Mass spectrometry indicated that the gel band contained multiple polypeptides, including DDB1 (Appendix A). DDB1 was one of the proteins with a high score in the mass spectrometry analysis, which indicated a greater abundance of the protein in the band. Since DDB1 is known to play an important role in replication stress and DNA damage, we selected DDB1 to further investigate its role in relation to NBS1.

To verify the specificity of the interaction between DDB1 and NBS1, we prepared recombinant proteins for pull-down assays. Initially, we generated recombinant NBS1-FLAG and DDB1-FLAG proteins in insect cells and conducted pull-down experiments using anti-FLAG antibody beads, which were either free of or loaded with NBS1-FLAG or DDB1-FLAG proteins. We incubated egg extracts with either NBS1-FLAG or DDB1-FLAG and subsequently analyzed them for the presence of DDB1 and NBS1. Our analysis revealed the presence of DDB1 in the NBS1 pull-down and vice versa (Figure 1B,C). To further validate the specificity of these interactions, we performed immunoprecipitation assays. We generated anti-DDB1 antibodies targeting a specific polypeptide fragment of DDB1. Egg extracts, either containing or lacking pA-pT, were subjected to immunoprecipitation using anti-NBS1 antibodies, followed by immunoblotting to detect DDB1 and TopBP1. We successfully identified DDB1 within the anti-NBS1 immunoprecipitates (Figure 1D). Notably, DDB1’s binding to NBS1 was not affected by the presence of pA-pT, while the interaction between NBS1 and TopBP1 was enhanced in the presence of pA-pT, consistent with previous findings [16,24].

We further investigated the interaction between NBS1 and DDB1 in human cells. Following transfection of HEK 293T cells with a vector coding for HA-tagged human DDB1 or a vector for FLAG-tagged human NBS1, we performed immunoprecipitation using anti-FLAG antibodies on cell lysates, subsequently analyzing the samples with anti-HA and anti-FLAG antibodies. After forty-eight hours post-transfection, the cells were subjected to IR treatment (10 Gy) to induce DSBs. We were able to detect HA-DDB1 in the anti-FLAG immunoprecipitates (Figure 1E). To assess the stoichiometry of the DDB1-NBS1 interaction, HEK 293T cells were transfected with decreasing amounts of the FLAG-tagged human NBS1 (pNBS1-FLAG) vector. We conducted anti-DDB1 immunoprecipitation from the lysates 48 h after transfection, followed by immunoblotting with anti-NBS1 antibodies. The association between NBS1 and DDB1 diminished in proportion to the decreasing expression levels of NBS1 (Appendix A). Additionally, after immunoprecipitating HA-DDB1 from cell lysates using anti-HA antibodies, NBS1 was identified (Figure 1F). Notably, the binding between DDB1 and NBS1 was independent of IR exposure, and we did not observe an increase in their interaction following replication stress induced by UV radiation (Appendix A). Collectively, these findings suggest that DDB1 specifically interacts with NBS1 in both Xenopus egg extracts and human cells.

### 2.2. DDB1 Associates with Several Proteins That Interact with NBS1, Including TopBP1 and MDC1, in Both Egg Extracts and Human Cells

TopBP1 is known to interact with the MRN complex in egg extracts, which includes the NBS1 subunit of the complex [16,24]. We set out to determine whether DDB1 can bind to TopBP1. To do this, we produced recombinant TopBP1-FLAG proteins in insect cells and incubated them with egg extracts, both in the presence and absence of pA-pT. Subsequently, we isolated the tagged proteins from the extracts and examined them for DDB1 and NBS1. Our results showed that DDB1 is associated with TopBP1 (Figure 2A) and the levels of DDB1 binding to TopBP1 were consistent regardless of the presence of pA-pT. As anticipated, NBS1 was also found to bind to TopBP1. We then investigated the interaction between TopBP1 and DDB1 in human cells. HEK 293T cells were either mock-treated or exposed to IR (10 Gy), with cell lysates prepared 2 h post-treatment. Control IgG immunoprecipitates and anti-DDB1 immunoprecipitates from these lysates were analyzed using anti-TopBP1 and anti-NBS1 antibodies. We observed the presence of TopBP1 and NBS1 in anti-DDB1 immunoprecipitates (Figure 2B). These results show that endogenous DDB1 interacts with both NBS1 and TopBP1. After transfection of HEK 293T cells with vectors expressing HA-DDB1 or FLAG-TopBP1, we performed immunoprecipitation with anti-FLAG antibodies on the cell lysates. This was followed by immunoblotting with anti-HA and anti-FLAG antibodies, which revealed HA-DDB1 in the anti-FLAG immunoprecipitates (Figure 2C). The binding of DDB1 to TopBP1 is independent of IR treatment. To further rule out the possibility that DDB1 associates with NBS1 and TopBP1 through secondary interactions with cellular DNA, we conducted immunoprecipitation experiments in the presence of DNase I. These results confirmed that DDB1 and TopBP1 interact specifically with NBS1 (Appendix A). Overall, our findings indicate that DDB1 specifically binds to TopBP1 in both egg extracts and human cells.

In our previous research, we established that MDC1 is a protein interacting with both TopBP1 and NBS1, and it is crucial for facilitating the association between TopBP1 and NBS1 in both egg extracts and human cells [24]. We proceeded to investigate whether DDB1 also binds to MDC1. To confirm these interactions, we conducted immunoprecipitation experiments using egg extracts either with or without pA-pT, using anti-MDC1 antibodies, and subsequently analyzed the immunoprecipitates for the presence of DDB1, NBS1, and TopBP1. Our results clearly showed the presence of DDB1, NBS1, and TopBP1 in the anti-MDC1 immunoprecipitates (Figure 3A), suggesting a specific interaction between DDB1 and MDC1 in egg extracts. Additionally, we examined the interactions of these proteins within the nuclear fractions of egg extracts. Egg extracts containing sperm nuclei were treated with or without EcoRI restriction endonuclease prior to preparing the nuclear fractions. EcoRI effectively generates DSBs in chromatin, triggering a robust checkpoint response in egg extracts [30,31]. We performed immunoprecipitation on the nuclear fractions using MDC1 antibodies conjugated to protein A magnetic beads and analyzed the samples with anti-DDB1, anti-NBS1, and anti-TopBP1 antibodies. As shown in Figure 3B, DDB1, NBS1, and TopBP1 were readily detected in the anti-MDC1 immunoprecipitates.

We also assessed whether DDB1 interacts with MDC1 in human cells. Following transfection of HEK 293T cells with a plasmid expressing HA-tagged human MDC1, we immunoprecipitated using anti-HA antibodies from the cell lysates and immunoblotted the resulting samples with anti-DDB1, anti-TopBP1, and anti-NBS1 antibodies. The analysis revealed the presence of DDB1, TopBP1, and NBS1 in the anti-HA immunoprecipitates (Figure 3C). Overall, these findings suggest that DDB1 specifically associates with several critical checkpoint regulators, including MDC1, TopBP1, and NBS1, in both egg extracts and human cells.

### 2.3. The Interaction Between NBS1 and DDB1 Is Mediated by MDC1

We then examined the role of MDC1 in facilitating the interaction between DDB1 and NBS1. Specifically, we sought to determine whether these interactions in egg extracts relied on the presence of MDC1. To conduct these experiments, we utilized anti-MDC1 antibodies to deplete endogenous MDC1 from the egg extracts as previously described [24]. As shown in Figure 3D, the depletion of MDC1 from the egg extracts was complete when using these antibodies. We conducted pull-down assays with anti-FLAG antibody beads, which either lacked or contained DDB1-FLAG, using extracts that were either depleted of MDC1 or mock-treated, both with and without pA-pT. Our analysis revealed the absence of interaction between DDB1 and NBS1 in extracts where MDC1 had been depleted (Figure 3E), suggesting that DDB1’s binding to NBS1 is contingent upon MDC1. These findings demonstrate that MDC1 plays a mediating role in the interaction between DDB1 and NBS1, similar to that between TopBP1 and NBS1 [24].

### 2.4. DDB1 Associates with the N-Terminal Region of MDC1

To explore the binding mechanism between NBS1 and DDB1 more thoroughly, we generated tagged fragments of NBS1 that encompassed the N-terminal (residues 1–410) and C-terminal (residues 338–762) portions of the protein. These fragments were incubated with anti-FLAG antibody beads in egg extracts. After isolating the tagged proteins with the beads, we found that DDB1 associated with the N-terminal fragment but not with the C-terminal part of NBS1 (Figure 4A). Similarly, we created N-terminal (residues 1–600) and C-terminal (residues 590–1140) fragments of DDB1 for incubation in egg extracts. We observed that NBS1 interacted with the C-terminal fragment (residues 590–1140) of DDB1, while there was no binding to the N-terminal fragment. These experiments indicated that the interaction between DDB1 and NBS1 is primarily mediated by the C-terminal region of DDB1 (Figure 4B).

To identify the specific region of MDC1 that interacts with DDB1, we conducted structure-function analyses to clarify which parts of MDC1 are involved in this binding. The N-terminal portion of MDC1 houses a single forkhead-associated (FHA) domain, while the C-terminal region contains two BRCT domains [19,32]. We generated tagged fragments of MDC1 covering the N-terminal (residues 1–1166) and C-terminal (residues 989–2100) sections of the protein and performed pull-down assays using anti-FLAG antibody beads with or without pA-pT. These fragments were incubated with the beads in egg extracts, and after isolating the tagged proteins, we found that DDB1 bound to the N-terminal fragment but not to the C-terminal fragment of MDC1 (Figure 4C). In previous research, we established that both NBS1 and TopBP1 bind to the N-terminal portion of MDC1 [24]. To further refine our understanding of the regions essential for DDB1 binding, we created various deletions within the N-terminal domain of MDC1. A GST-tagged fragment corresponding to residues 1–350 of MDC1 demonstrated the ability to bind to DDB1 in egg extracts (Figure 4D). To investigate which region of DDB1 interacts with MDC1, we constructed GST-MDC1(1–130), GST-MDC1(110–260), and GST-MDC1(250–350) fragments and incubated them in egg extracts with recombinant DDB1-FLAG on anti-FLAG antibody beads, both in the presence and absence of pA-pT. Following the isolation of the beads, we performed immunoblotting for GST. We detected the binding of the GST-MDC1(250–350) fragment to DDB1-FLAG beads (Figure 4E). These findings suggest that residues 250–350 of MDC1 are critical for interacting with DDB1. Our earlier studies showed that residues 490–600 of MDC1 are involved in binding with NBS1 in egg extracts [24], specifically through the SDT repeats located in that region [20,21,22,23]. Furthermore, MDC1 is known to associate with the tandem BRCT repeats of NBS1 [24]. Overall, these results support the conclusion that MDC1 serves as a connector between DDB1 and NBS1 (Figure 5E).

### 2.5. DDB1 Associates with Damaged Chromatin in Egg Extracts

Given that DDB1 engages with several NBS1-associated proteins involved in DNA damage responses, we investigated whether DDB1 binds to damaged chromatin. To explore this, we incubated Xenopus sperm chromatin in egg extracts with or without EcoRI to induce double-stranded breaks. We also treated the extracts with aphidicolin to cause stalled DNA replication forks. Afterward, we prepared nuclear and chromatin fractions to analyze the levels of DDB1 and other interacting checkpoint proteins such as MDC1, TopBP1, NBS1, and Orc2 (Figure 5A). Our observations indicated that DDB1 exhibited greater association with chromatin damaged by EcoRI and aphidicolin compared to undamaged chromatin (Figure 5A). Similar increased binding was noted for MDC1, TopBP1, and NBS1 in the presence of both EcoRI and aphidicolin, consistent with previous findings. Overall, these findings suggest that DDB1 forms a complex with other interacting proteins, and this complex shows significantly enhanced binding to damaged chromatin (Figure 5A,E).

### 2.6. The Role of DDB1 in Regulation of DNA Damage Checkpoint Signaling

We investigated the function of DDB1 in the context of DNA damage checkpoint signaling. Specifically, we assessed the activation of Chk1 in both the presence and absence of DDB1 following DNA damage. To do this, endogenous DDB1 was depleted from egg extracts using anti-DDB1 antibodies, and the extracts were then incubated with Xenopus sperm chromatin. This was done in both mock-depleted and DDB1-depleted extracts, with or without EcoRI or aphidicolin. Nuclear fractions were isolated and analyzed via immunoblotting using antibodies specific for Xenopus DDB1, phospho-Ser-344 of Chk1, as well as Xenopus Chk1 and Chk2. As shown in Figure 5B, we detected robust phosphorylation of Chk1 at S344 in mock-treated extracts in response to both double-strand breaks (DSBs) and replication block damage. Notably, this phosphorylation level was slightly elevated in the DDB1-depleted extracts. These findings indicate that the absence of DDB1 resulted in increased phosphorylation of Chk1 in the presence of both DSBs and replication block damage.

To evaluate the functional significance of DDB1 in checkpoint signaling, we implemented a siRNA-based approach in human cells. HeLa cells were treated with siRNA targeting DDB1, followed by exposure to IR or UV to induce double-strand breaks (DSBs) or replication block damage, respectively. The treatment with DDB1 siRNA resulted in a significant reduction of DDB1 levels, whereas control siRNA did not impact DDB1 expression under the same experimental conditions. After transfecting HEK 293T cells with either control or DDB1 siRNA for three days, we first examined whether the depletion of DDB1 influenced cell cycle progression in response to DSBs. In the control cells, we observed that S and G2 phase cells were arrested following IR treatment (Appendix A). Similarly, DDB1-depleted cells also exhibited cell cycle arrest when subjected to IR (Appendix A). These findings indicate that the depletion of DDB1 does not alleviate the IR-induced cell cycle arrest. In the control cells, we detected phosphorylation of Chk1 at S317 in response to IR, which was found to be slightly enhanced in DDB1-depleted cells (Figure 5C,D). Additionally, we observed the phosphorylation of ATM, TopBP1, and γH2AX in both control and DDB1-depleted cells following IR treatment (Figure 5C). These results demonstrate that the ATM pathway for DSB checkpoint activation remains well-functioning in DDB1-depleted cells after exposure to IR. Enhanced phosphorylation of Chk1 was also noted in DDB1-depleted cells treated with UV (Appendix A). Collectively, these findings suggest that DDB1 plays a role in facilitating checkpoint recovery from both DSBs and replication block damage.

## 3. Discussion

This study established DDB1 as an interacting partner of NBS1 through an exploration of its function and regulation in both Xenopus egg extracts and human cells. To thoroughly investigate the entire genome, it is essential to identify all types of DNA structural abnormalities, including DNA nicks, gaps, double-strand breaks (DSBs), and replication block damage. There are at least five distinct pathways involving protein complexes that detect and signal various forms of DNA damage [33]. Our focus was directed towards the ATM and ATR pathways. ATM is primarily linked to DSBs, while ATR is predominantly involved in responses to replication block damage. Many of these proteins play critical roles within the DNA damage response (DDR). The MRN complex is vital during the initial phases of the cellular reaction to DSBs [34,35,36]. It directly associates with the ends of broken DNA, processes the DNA structures, and modulates checkpoint signaling. The MRN complex facilitates the resection of broken DNA ends, thereby enhancing checkpoint signaling by producing single-stranded DNA near the sites of DNA breaks. Additionally, the MRN complex is essential for the activation of ATM in response to DSBs and also aids in the activation of ATR-ATRIP when DSBs occur [7,8,16,37].

TopBP1 is responsible for the direct activation of the ATR-ATRIP complex [14]. Additionally, it serves as a link between ATM and ATR, particularly in response to DSBs [9]. ATM phosphorylates TopBP1 at the S1131 site, significantly enhancing TopBP1’s ability to activate ATR. This phosphorylation at S1131 is essential solely for checkpoint responses to DSBs and is not required for reactions to replication damage. The MRN complex is crucial for facilitating the recruitment of ATM to TopBP1 in response to DSBs [16]. TopBP1 associates with the MRN complex through its interaction with the NBS1 component [16,17]. Recent research has indicated that MDC1 plays a mediating role in the interaction between TopBP1 and NBS1 within the MRN complex [24].

This study demonstrated that DDB1 interacts with NBS1 in both egg extracts and human cells. While the interaction between NBS1 and TopBP1 is significantly enhanced in response to DSBs in egg extracts [16,24], the interaction between NBS1 and DDB1 occurs in a manner that is independent of damage, in both egg extracts and human cells. Pull-down and immunoprecipitation assays confirmed that NBS1 and DDB1 physically associate, with the N-terminal region of NBS1 binding to the C-terminal region of DDB1. The N-terminal portion of NBS1 contains one FHA domain and two BRCT domains, whereas the C-terminal region of DDB1 does not possess any known unique domains.

In addition to DDB1’s interaction with NBS1, DDB1 also associates with TopBP1 and MDC1 in both egg extracts and human cells. Our previous research indicated that TopBP1 and NBS1 interact with the N-terminal segment of MDC1. This N-terminal portion contains one forkhead-associated domain (residues 46–96), while the C-terminal section consists of two BRCT domains. By employing various deletions within the N-terminal domain of MDC1, we discovered that the removal of residues 161–230 notably diminished its ability to interact with TopBP1 [24]. A fragment of MDC1 that includes residues 161–230 was capable of binding to TopBP1 in egg extracts. Thus, it can be concluded that residues 161–230 of MDC1 are essential for TopBP1 binding [24]. Prior studies have shown that TopBP1 interacts directly with the Ser-Asp-Thr (SDT) repeats located in the N-terminal domain of MDC1 [38], although the residues 161–230 of Xenopus MDC1 do not contain these SDT repeats. NBS1 forms an interaction with residues 490–600 of MDC1 in egg extracts, which aligns with findings suggesting that the SDT repeats present in this region are crucial for its association with NBS1 [20,21,22,23]. Furthermore, the phosphorylation of conserved SDT repeats at the N-terminus of MDC1 promotes the recruitment of NBS1 to sites of DNA damage [20,21,22,23]. In egg extracts lacking MDC1, we did not observe any interaction between TopBP1 and NBS1, suggesting that MDC1 serves as a critical link between TopBP1 and the MRN complex.

In this study, we also discovered that DDB1 interacts with the N-terminal region of MDC1 in egg extracts. Our analysis revealed that a fragment consisting of residues 250–350 of MDC1 was capable of binding to DDB1, based on various deletion constructs of MDC1’s N-terminal region (Figure 4). Additionally, our binding experiments with the N- and C-terminal halves of NBS1 indicated that the N-terminal region of NBS1 is essential for its interaction with DDB1. This N-terminal segment of NBS1 comprises tandem BRCT domains that are necessary for its interaction with MDC1. For NBS1 to effectively activate Chk1 in response to DSBs, it must possess intact BRCT domains [16]. Notably, the interaction between DDB1 and NBS1 was significantly diminished in MDC1-depleted egg extracts (Figure 3E). These findings suggest that MDC1 serves as a mediator in the interaction between DDB1 and NBS1, utilizing a mechanism akin to that observed between TopBP1 and NBS1. The interactions of these proteins are depicted in Figure 5E.

DDB1 plays a role in the repair of damage caused by replication blocks. In both normal cells and those under replication stress, DDB1 directs Chk1 to the Cul4 E3 ligase complex. The Cul4A/DDB1 complex is responsible for ubiquitinating Chk1, thereby regulating its abundance during replication stress, and is thought to play a significant role in the recovery process from such stresses and DNA damage [26]. Additionally, DDB1 ubiquitinate Cdt1, the replication licensing protein in response to DNA damage caused by UV radiation, resulting in its rapid degradation. This process inhibits the replication of damaged DNA, thereby facilitating DNA repair and helping to maintain genomic integrity [39,40]. A recent study has shown that the Cul4A-DDB1 E3 ligase ubiquitinates the CDK inhibitor p21, with this ubiquitination depending on its association with proliferating cell nuclear antigen (PCNA), constituting an additional mechanism that contributes to DNA repair [41]. In this study, we found no reduction in the protein levels of NBS1, TopBP1, or MDC1 in either normal or damaged egg extracts. To evaluate DDB1’s role at the DNA damage checkpoint, we assessed the phosphorylation of Chk1 in response to DSBs and replication block damage in egg extracts depleted of DDB1. We observed a significant increase in Chk1 phosphorylation following both EcoRI and aphidicolin treatments. Furthermore, we utilized siRNAs to knock down DDB1 in human cells. In cells treated with control siRNA, we noted an increase in Chk1 phosphorylation in response to IR or UV exposure. Remarkably, DDB1-depleted cells exhibited even higher levels of Chk1 phosphorylation compared to control cells. These findings support the notion that DDB1 is essential for efficient checkpoint recovery from DSBs and replicative stress.

In conclusion, we have identified DDB1 as an interacting partner of NBS1 in both egg extracts and human cells. Additionally, our findings indicate that MDC1 facilitates the interaction between DDB1 and NBS1. The depletion of DDB1 results in increased activation of Chk1 in response to DNA damage. These results strongly imply that DDB1 plays a significant role in the recovery of checkpoints following DNA damage.

## 4. Materials and Methods

### 4.1. Preparation of Xenopus Egg Extracts and Cell Culture

Xenopus egg extracts were prepared following previously established protocols [42]. To initiate checkpoint responses, the extracts were treated with 50 μg/mL pA-pT. DSBs were induced by adding 0.05 U/μL of the restriction enzyme EcoRI to the egg extracts. Activation of the DNA replication checkpoint was achieved by treating the extracts with 100 μg/mL of aphidicolin, as previously described [30]. All cell lines were cultured in Dulbecco’s Modified Eagle Medium (DMEM), supplemented with 10% fetal bovine serum (FBS), 100 U/mL penicillin, and 100 μg/mL streptomycin. Human cell lines, including HEK 293T and HeLa, were sourced from the American Type Culture Collection (ATCC, Manassas, VA, USA). Regular testing for mycoplasma contamination was performed, and the cell lines were authenticated through STR analysis. Insect cells, specifically Sf9 cells, were cultured in Grace’s insect medium enriched with 3330 mg/L of lactalbumin hydrolysate and 3330 mg/L of yeastolate (Gibco, Waltham, MA, USA) and used for Baculoviral protein expression.

### 4.2. Pulldown of Recombinant Xenopus NBS1 from Xenopus Egg Extracts

Recombinant FLAG-tagged Xenopus NBS1 (NBS1-F) was incubated with egg extracts containing pA-pT and subsequently isolated using anti-FLAG antibodies coupled to protein G magnetic beads [16]. Following the retrieval of the beads and separation via SDS-PAGE, the relevant experimental lane was excised and subjected to mass spectrometry analysis, in accordance with previously established procedures [43]. Appendix A presents a list of proteins identified through affinity purification and LC-MS/MS analysis.

### 4.3. Antibodies

Full-length cDNA clones for both Xenopus DDB1 and human DDB1 were amplified from a Xenopus laevis oocyte cDNA library and a human placenta cDNA library, respectively, using PCR. The resulting DDB1 PCR products were confirmed through Sanger sequencing. A DNA fragment encoding the amino acids 700–1140 of Xenopus DDB1 was generated by PCR and subsequently cloned into the pET-His6 expression vector. The His6-DDB1(700–1140) protein was expressed in *Escherichia coli*, purified using nickel agarose, and used to generate rabbit antibodies at a commercial facility. The affinity-purified DDB1 antibodies could detect both Xenopus and human DDB1 proteins in immunoblotting assays. Additionally, affinity-purified antibodies against Xenopus NBS1, TopBP1, and MDC1 were described in previous studies [14,16,24]. Primary antibodies were sourced from established commercial vendors, including HA (Santa Cruz Biotechnology, Dallas, TX, USA), FLAG (Sigma-Aldrich, St. Louis, MO, USA), human NBS1 (Cell Signaling Technology, Danvers, MA, USA), human β-Actin (AbFrontier, Seoul, South Korea), human α-Tubulin (Santa Cruz Biotechnology, Dallas, TX, USA), human ATM (Cell Signaling Technology, Danvers, MA, USA), human ATM phospho-Ser1981 (Cell Signaling Technology, Danvers, MA, USA), human TopBP1 (Bethyl Laboratories, Montgomery, TX, USA), human TopBP1 phospho-Ser1138 (Abcam, Cambridge, UK), human Chk1 (Santa Cruz Biotechnology, Dallas, TX, USA), human Chk1 phospho-Ser317 (Cell Signaling Technology, Danvers, MA, USA), human Chk2 (Santa Cruz Biotechnology, Dallas, TX, USA), human Chk2 phospho-Thr68 (Cell Signaling Technology, Danvers, MA, USA), and human MDC1 (Cell Signaling Technology, Danvers, MA, USA).

### 4.4. Immunoprecipitation and Immunodepletion

For the immunoprecipitation assays, 100 μL of egg extracts were mixed with Affiprep-protein A beads (Bio-Rad, Hercules, CA, USA) that had been coated with 3 μg of either anti-DDB1, anti-NBS1, anti-TopBP1, or anti-MDC1 antibodies, and were incubated with constant agitation at 4 °C for 45 min. The immunoprecipitation was performed according to previously established protocols [16,44]. To deplete DDB1, 100 μL of interphase extracts were incubated with 40 μg of anti-DDB1 antibodies attached to 20 μL of Affiprep-protein A beads at 4 °C for 45 min. For mock depletion, the same quantity of control rabbit IgG was utilized. After the incubation period, the beads were removed via centrifugation and the supernatants underwent a second round of depletion. MDC1 was immunodepleted following the methods described in earlier studies [24]. For immunoprecipitation in human cells, HEK 293T cells were transfected with the specified plasmids for 24 h before being either treated with ionizing radiation (10 Gy) or left untreated. The cells were collected and lysed 1 h after IR treatment using a lysis buffer composed of 50 mM Tris, 150 mM NaCl, 1 mM EDTA, 0.5% NP-40, 1% Triton X-100, supplemented with protease inhibitors. The lysates were kept on ice for 30 min, followed by centrifugation at 13,000 rpm at 4 °C to collect the supernatants. Whole cell lysates were then incubated with 2 μg of the relevant antibody linked to Dynabeads protein G for 3 h at 4 °C with rotation, followed by three washes with the lysis buffer. The proteins bound to the beads were subsequently analyzed by immunoblotting using the appropriate antibodies.

### 4.5. Production of Recombinant Proteins

DNA fragments encoding full-length DDB1, as well as segments consisting of residues 1–600 and 590–1140 of DDB1, were synthesized using PCR and then inserted into the pFastBac-FLAG vector, allowing for the expression of FLAG-tagged proteins at the C-terminus. Recombinant baculoviruses were produced using the Bac-to-Bac system (Invitrogen). Additionally, we generated recombinant full-length HF-TopBP1, which features both hemagglutinin (HA) and His6 tags at the N-terminus, along with a FLAG tag at the C-terminus, as well as NBS1 and MDC1 proteins, both tagged with FLAG at the C-terminus, following previously established methods [16,24]. The pcDNA3.1 vector was utilized to express FLAG-tagged human DDB1, NBS1, TopBP1, and MDC1 in human cell lines.

### 4.6. Preparation of Nuclear and Chromatin Fractions

Egg extracts (100–200 μL) containing sperm nuclei at a concentration of 3000 nuclei/μL were incubated under specified conditions. The isolation of nuclear and chromatin fractions was carried out as described [45].

### 4.7. Transfection of Plasmid and Small Interfering RNAs

Cells were transfected with plasmid DNA utilizing Lipofectamine 3000 (Invitrogen, Thermo Fisher Scientific, Waltham, MA, USA) and small interfering RNAs (siRNAs) were delivered using Lipofectamine RNAiMax Reagent (Invitrogen, Thermo Fisher Scientific, Waltham, MA, USA) following the manufacturer’s instructions. Two siRNA duplexes targeting human DDB1 were obtained from Bioneer, with the sense strand sequences 5′-CACAUGAUUCCAGCCAUCA-3′ and 5′-GAGAUGGAGCGCUUUUCUA-3′.

## Figures and Tables

**Figure 1 ijms-25-13097-f001:**
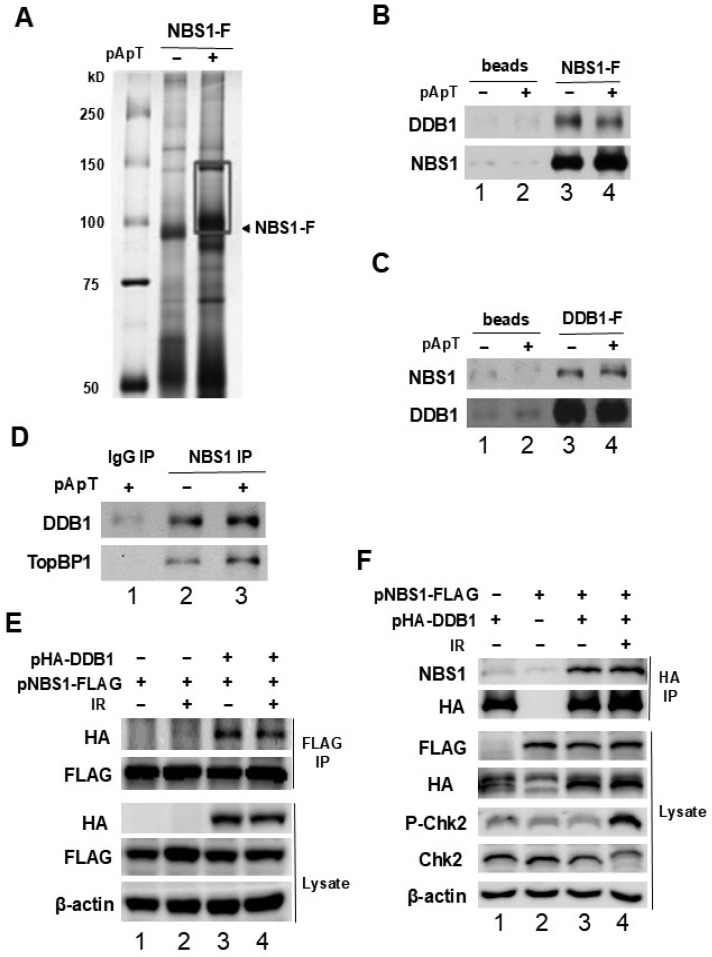
The association of DDB1 with NBS1 in Xenopus egg extracts and human cells. (**A**) Interphase egg extracts containing NBS1-FLAG, with or without pA-pT, were subjected to immunoprecipitation using anti-FLAG antibodies. The resulting immunoprecipitates underwent SDS-PAGE followed by silver staining. The boxed region of the gel was further analyzed by mass spectrometry. Lane 1 displays molecular mass markers. (**B**) Anti-FLAG antibody beads lacking recombinant proteins (lanes 1 and 2) and those containing NBS1-FLAG (lanes 3 and 4) were incubated in egg extracts with or without pA-pT. The beads were collected and analyzed via immunoblotting using anti-DDB1 and anti-NBS1 antibodies. (**C**) Anti-FLAG antibody beads devoid of recombinant proteins (lanes 1 and 2) and those with DDB1-FLAG (lanes 3 and 4) were incubated in egg extracts with or without pA-pT. After isolation, the beads were immunoblotted with anti-NBS1 and anti-DDB1 antibodies. (**D**) Control immunoprecipitates and anti-NBS1 IPs from egg extracts, with or without pA-pT, were analyzed by immunoblotting for DDB1 and TopBP1. (**E**) HEK 293T cells were transfected with either a control plasmid or vectors that express HA-tagged human DDB1 (HA-DDB1) and FLAG-tagged human NBS1 (NBS1-FLAG). Following a 48 h incubation, the cells received either mock treatment or were exposed to IR (10 Gy). After 2 h, cell lysates were collected and anti-FLAG immunoprecipitations were performed and subsequently analyzed through immunoblotting using anti-HA and anti-FLAG antibodies, while the cell lysates were examined with the specified antibodies. (**F**) HEK 293T cells were transfected with either a control vector or vectors expressing FLAG-tagged human NBS1 (NBS1-FLAG) and HA-tagged human DDB1 (HA-DDB1). Cell lysates were prepared as described in (**E**). Anti-HA IP from these lysates was immunoblotted for the specified proteins.

**Figure 2 ijms-25-13097-f002:**
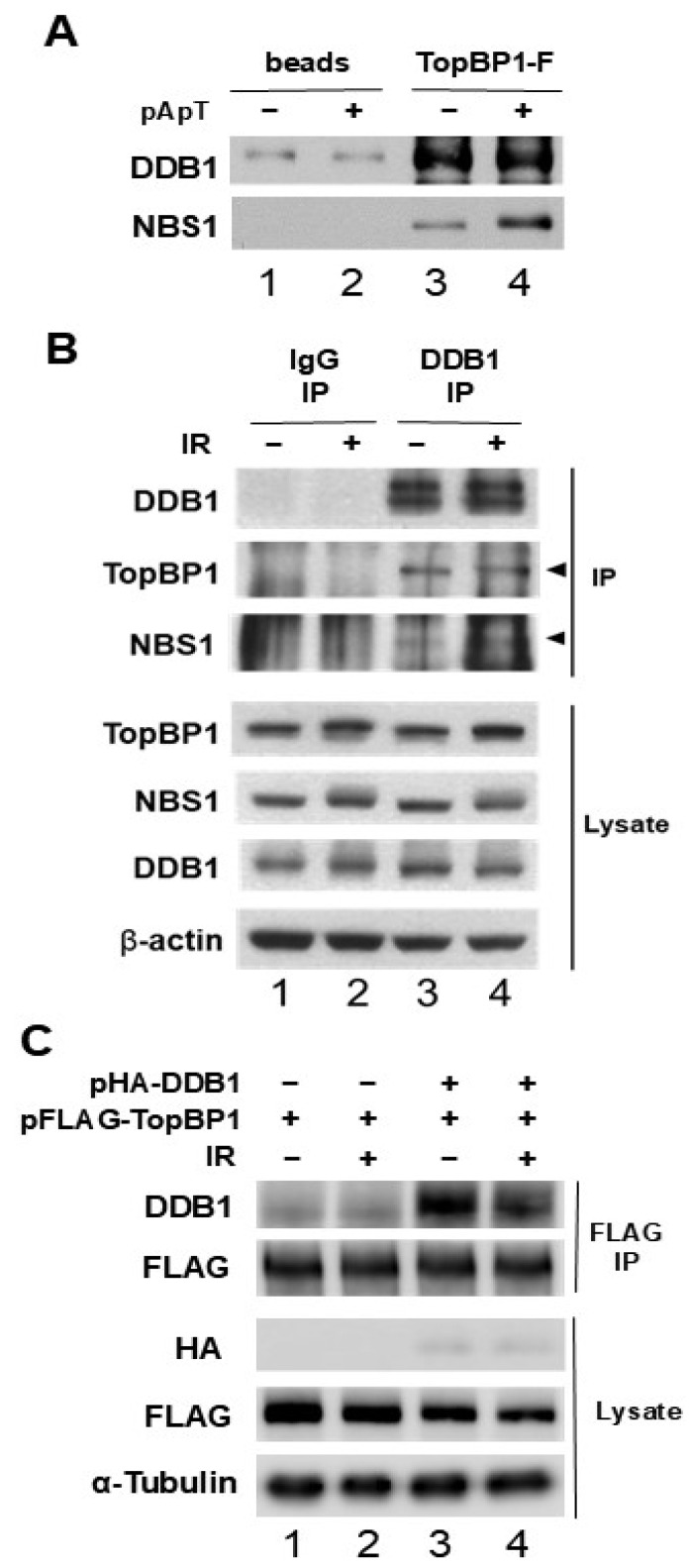
DDB1 interaction with TopBP1 in Xenopus egg extracts and human cells. (**A**) Anti-FLAG antibody beads lacking recombinant proteins (lanes 1 and 2) and those containing TopBP1-FLAG (lanes 3 and 4) were incubated with egg extracts, both with and without pA-pT. The beads were collected and subjected to immunoblotting using anti-DDB1 and anti-NBS1 antibodies. (**B**) HEK 293T cells were either mock-treated or subjected to IR (10 Gy). At 2 h after treatment, cell lysates were obtained. Immunoprecipitation with control IgG and anti-DDB1 from these lysates was followed by immunoblotting with anti-DDB1, anti-TopBP1, and anti-NBS1 antibodies. The corresponding antibodies were used to analyze the cell lysates as well. (**C**) HEK 293T cells were transfected with either a control plasmid or vectors that express HA-tagged human DDB1 (HA-DDB1) and FLAG-tagged human TopBP1 (FLAG-TopBP1). Cell lysates were prepared as outlined in (**B**). Immunoprecipitation with anti-FLAG antibodies from the cell lysates was subsequently analyzed by immunoblotting for the specified proteins.

**Figure 3 ijms-25-13097-f003:**
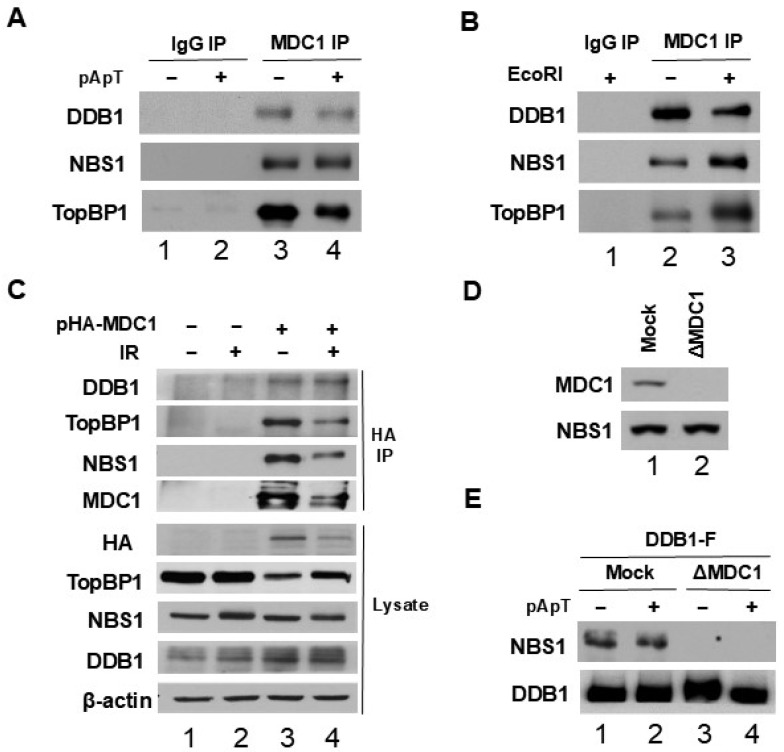
MDC1 mediates the interaction between NBS1 and DDB1. (**A**) Immunoprecipitates (IP) were obtained using control antibodies (lanes 1 and 2) and anti-MDC1 antibodies (lanes 3 and 4) from egg extracts, with or without pA-pT, and were subsequently analyzed by immunoblotting using anti-DDB1, anti-NBS1, and anti-TopBP1 antibodies. (**B**) Egg extracts containing sperm nuclei were incubated for 60 min either without EcoRI (lane 2) or in the presence of EcoRI (lanes 1 and 3). After this incubation, nuclear fractions were prepared and treated with protein A magnetic beads containing either control antibodies (lane 1) or anti-MDC1 antibodies (lanes 2 and 3). The beads were collected and analyzed via immunoblotting for DDB1, NBS1, and TopBP1. (**C**) HEK 293T cells were transfected with a control plasmid or a plasmid that expresses HA-tagged human MDC1 (HA-MDC1). After 48 h, the cells received either mock treatment or were subjected to IR at 10 Gy. Two hours later, cell extracts were prepared, followed by anti-HA immunoprecipitation and immunoblotting with antibodies against DDB1, TopBP1, NBS1, and MDC1. The cell lysates were also analyzed with specified antibodies. (**D**) Egg extracts were either mock-depleted using control antibodies (lane 1) or subjected to immunodepletion with anti-MDC1 antibodies (lane 2). Following this, the extracts were processed for immunoblotting using antibodies against MDC1 and NBS1. (**E**) Egg extracts that were either mock-depleted (lanes 1 and 2) or depleted of MDC1 (lanes 3 and 4), with or without the presence of pA-pT, were incubated with anti-FLAG antibody beads containing DDB1-FLAG. The beads were then collected again and subjected to immunoblotting using anti-NBS1 and anti-DDB1 antibodies.

**Figure 4 ijms-25-13097-f004:**
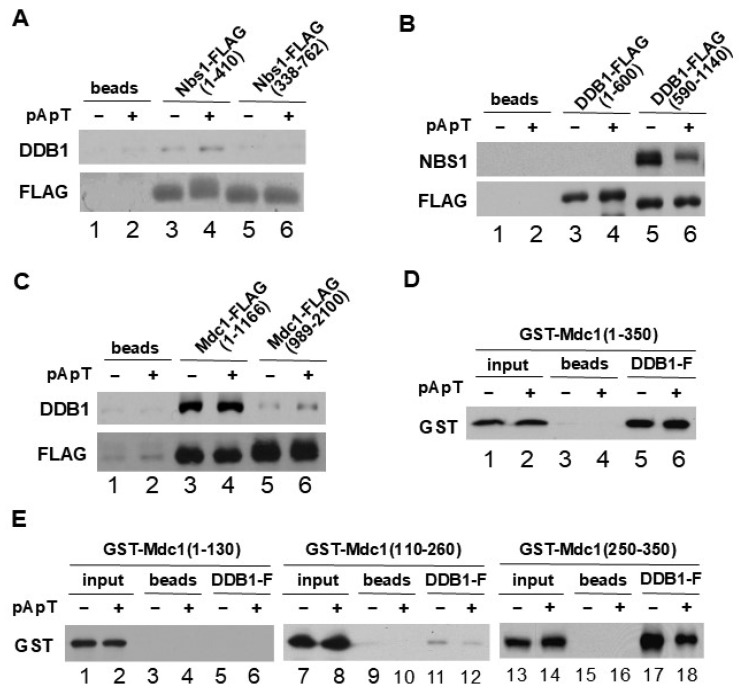
Characterization of DDB1 binding sites on NBS1 and MDC1. (**A**) Anti-FLAG antibody beads that included no recombinant protein (lanes 1 and 2), NBS1-FLAG(1–410) (lanes 3 and 4), or NBS1-FLAG(338–762) (lanes 5 and 6) were incubated in egg extracts with or without of pA-pT. After isolating the beads, they were analyzed by immunoblotting with anti-DDB1 and anti-FLAG antibodies. (**B**) Anti-FLAG antibody beads that were either free of recombinant protein or contained DDB1-FLAG(1–600) or DDB1-FLAG(590–1140) were incubated in egg extracts, both in the presence and absence of pA-pT. After incubation, the beads were collected again and analyzed by immunoblotting with anti-NBS1 and anti-FLAG antibodies. (**C**) Anti-FLAG antibody beads that lacked recombinant proteins (lanes 1 and 2) or contained MDC1-FLAG(1–1166) (lanes 3 and 4) and MDC1-FLAG(989–2100) (lanes 5 and 6) were incubated in egg extracts with or without pA-pT. The beads were collected and immunoblotted using anti-DDB1 and anti-FLAG antibodies. (**D**) Recombinant DDB1-FLAG was immobilized on anti-FLAG antibody beads (lanes 5 and 6) and incubated with egg extracts containing GST-MDC1(1–350) in both the presence and absence of pA-pT. The beads were re-isolated and analyzed for GST by immunoblotting. (**E**) Recombinant DDB1-FLAG on anti-FLAG antibody beads were incubated with egg extracts containing GST-MDC1(1–130), GST-MDC1(110–260), or GST-MDC1(250–350) with or without pA-pT. After retrieval, the beads were immunoblotted with anti-GST antibodies.

**Figure 5 ijms-25-13097-f005:**
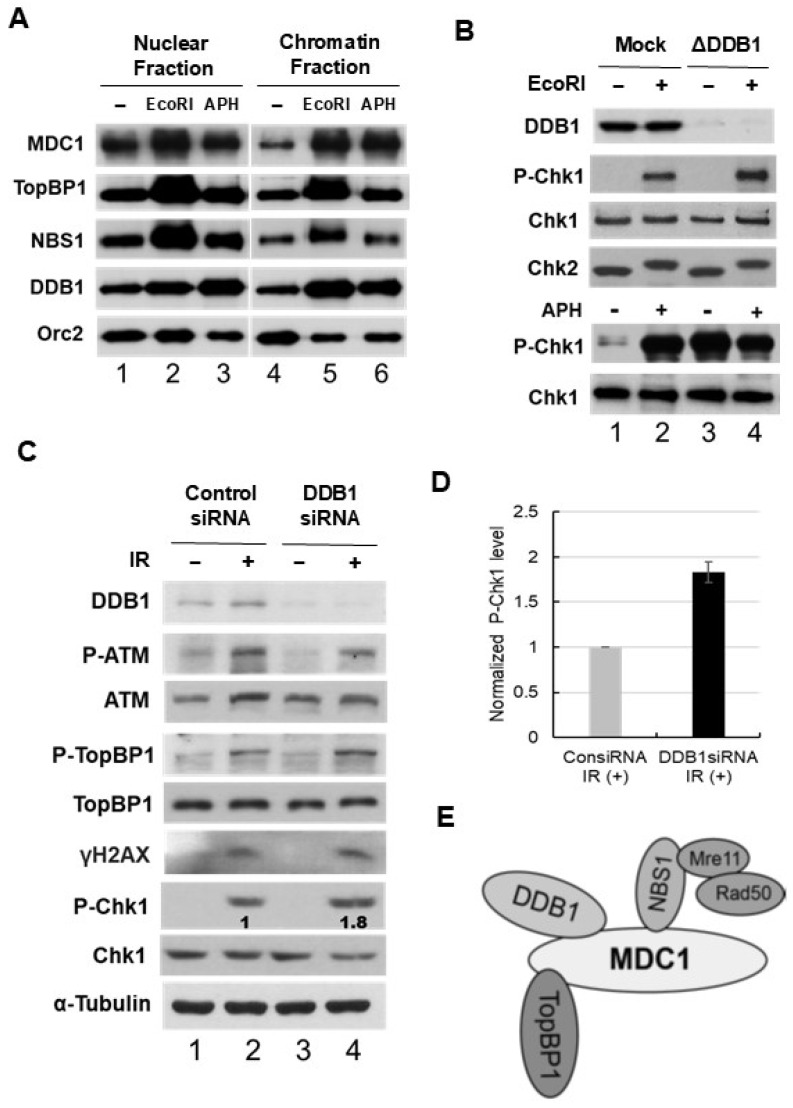
The role of DDB1 in regulating Chk1 activation in response to DNA damage. (**A**) Egg extracts containing sperm nuclei were treated under three conditions: without any additives (lanes 1 and 4), with 0.05 U/μL EcoRI (lanes 2 and 5), or with 50 μg/mL aphidicolin (Aph) (lanes 3 and 6). The nuclear fractions (lanes 1–3) and chromatin fractions (lanes 4–6) were analyzed by SDS-PAGE followed by immunoblotting for MDC1, TopBP1, NBS1, DDB1, and Orc2. EcoRI induces double-stranded DNA breaks, while aphidicolin causes DNA replication blockage. (**B**) Egg extracts were treated with control antibodies for mock depletion (lanes 1 and 2) or subjected to immunodepletion using anti-DDB1 antibodies (lanes 3 and 4). Both mock-depleted and DDB1-depleted extracts containing sperm nuclei were incubated for 90 min, either without treatment or with EcoRI or aphidicolin (APH). Subsequently, nuclear fractions were isolated and immunoblotted with antibodies specific for Xenopus DDB1, phospho-Ser-344 of Chk1, Xenopus Chk1, and Xenopus Chk2. (**C**) HeLa cells were transfected with either control siRNA or DDB1 siRNA1. They were then either mock-treated (lanes 1 and 3) or exposed to IR (10 Gy) (lanes 2 and 4). Two hours post-treatment, cell lysates were prepared and analyzed via immunoblotting with the specified antibodies. The phospho-Chk1 signals (lane 4) were quantified and normalized against those in lane 2, with results displayed beneath each band. (**D**) The graph illustrates the densitometric analysis results from the immunoblots, showing normalized phospho-Chk1 levels in DDB1 siRNA-treated cells compared to control siRNA-treated cells. The data come from four independent experiments and are presented as mean ± SD. (**E**) Proposed model for the interactions among MDC1, TopBP1, DDB1, and the MRN complex. MDC1 facilitates the connection between DDB1 and NBS1 within the MRN complex.

## Data Availability

Data is contained within the article and Appendix A.

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
