# Peer review of "Interaction of DDB1 with NBS1 in a DNA Damage Checkpoint Pathway"

_ijms, 2024, doi:10.3390/ijms252313097_

Round 1
Reviewer 1 Report
Comments and Suggestions for Authors
In this paper, the authors analyzed NBS1-binding proteins in Xenopus egg extract using mass spectrometry and found that DDB1 is associated with NBS1 in both the presence and absence of DSB (pA-pT) in the extract, as well as in human cells. Overall, it is a well-organized research paper, and most conclusions are supported by experimental data. However, some concerns listed below should be addressed before this paper is ready for publication.
- The rationale behind the authors' specific selection of proteins sized 90-150 kDa for mass spectrometry analysis in Figure 1A should be addressed in the main text.
- The reason for selecting DDB1 among other NBS1-interacting partners should be clarified in the main text.
- The physiological consequences of the DDB1-NBS1 interaction, specifically CHK1 activation, are not presented as a main dataset. The physiological importance of this interaction, such as its role in apoptosis, cell cycle arrest, or cellular senescence, should be demonstrated with experimental data.
In this paper, the authors analyzed NBS1-binding proteins in Xenopus egg extract using mass spectrometry and found that DDB1 is associated with NBS1 in both the presence and absence of DSB (pA-pT) in the extract, as well as in human cells. Overall, it is a well-organized research paper, and most conclusions are supported by experimental data. However, some concerns listed below should be addressed before this paper is ready for publication.
- The rationale behind the authors' specific selection of proteins sized 90-150 kDa for mass spectrometry analysis in Figure 1A should be addressed in the main text.
- The reason for selecting DDB1 among other NBS1-interacting partners should be clarified in the main text.
- The physiological consequences of the DDB1-NBS1 interaction, specifically CHK1 activation, are not presented as a main dataset. The physiological importance of this interaction, such as its role in apoptosis, cell cycle arrest, or cellular senescence, should be demonstrated with experimental data.
Reviewer 2 Report
Comments and Suggestions for Authors
In the manuscript entitled “Interaction of DDB1 with NBS1 in a DNA damage checkpoint pathway” the authors found that DNA damage- binding protein1 (DDB1) is a molecular partner of NBS1 ().
In this study, the authors performed pull-down and immunoprecipitation experiments by using either Xenopus egg extracts and human cells; in both cellular systems, it was confirmed the specific interaction between NBS1 and DDB1. Further evidence indicated that MDC1 () is also required for the specific binding of DDB1 and NBS1, as observed in experiments done with Xenopus egg extracts lacking of MDC1. Lastly, the authors observed that DDB1 depletion promotes Chk1 activation in response to DNA damage, suggesting that DDB1 (and probably its binding with NBS1) may regulate some aspects of the DNA damage response.
The paper is interesting, with major revisions, it will find its place in IJMS.
In this manuscript, the authors found some specific binding with DDB1 and specific protein complex involved in DDR. However, it will be important to validate the occurrence of such protein interactions within a cellular context. Could the authors reinforce the data by performing experiments of Immunofluorescence using tagged- constructs?
Fig. 2 panel B. About the NBS1 staining, it is clear that IR does promote some post-translational modifications on NBS1, could the authors make a comment about this? This is also evident in fig.3 panel C. Did the authors checked NBS1 staining in control siRNA and in DDB1siRNA2 cells exposed to IR?
Fig. 5 panel B. It is clear that following ECORI treatment Chk2 showed a sharp mobility shift. Could the authors add to this panel P-chk2 staining, as well?
Fig. S4 It will be nice to add to this figure either some western blot analysis or even IF data to follow the effect of IR in the same cells subjected to flow cytometry analysis.
Fig. S5 panel a. Could the authors also check the expression of NSB1 either in Control siRNA and in DDB1siRNA2 cells?
The text of the manuscript could be significantly improved because it is not easy to read.
Round 2
Reviewer 1 Report
Comments and Suggestions for Authors
The authors have appropriately addressed all concerns during the revision. I have no further comments and recommend accepting the paper for publication.